# Origin and Evolution of Ore-Forming Fluids at the Small-Sized Gold Deposits in the Khudolaz Area, Southern Urals

Ildar R. Rakhimov [1,*], Natalia N. Ankusheva [2], Aidar A. Samigullin [1] and Svetlana N. Shanina [3]

1    Ufa Federal Research Center, Institute of Geology, Russian Academy of Sciences, 16/2 Karl Marx Street, 450077 Ufa, Russia; ig@ufaras.ru
2    South Urals Federal Research Center of Mineralogy and Geoecology of UB RAS, Ilmeny Reserve, 456300 Miass, Russia; ankusheva@mail.ru
3    Institute of Geology of Komi Science Centre of the Ural Branch of RAS, 54, Pervomayskaya st., 167982 Syktyvkar, Russia; shanina@geo.komisc.ru
*    Correspondence: rigel92@mail.ru; Tel.: +7-919-159-0904

**Abstract:** Lode gold deposits are widespread in orogenic belts of various ages and are a valuable gold source, but their genesis remains debatable. The close relationship between native gold and quartz was considered a reason to search for acid-magmatic sources of heat and fluids (i.e., granite intrusions), while small gabbro bodies were often ignored. Six minor gold deposits associated with NE-strike faults were studied in the Khudolaz area of the South Urals (Tukan, Bilyan-Tau, Fazly-Tau, Muildy-Tamak, Alasiya-II and Isyanbet-I). It was established, for the first time, that all of the studied deposits are similar geologically but differ in mineralogical diversity of ore-bearing quartz veins, which is due to the different composition of host rocks and ore-bearing intrusions of the Khudolaz (325–329 Ma, U-Pb) and the Ulugurtau (321 ± 15 Ma, Sm-Nd) ultramafic-mafic complexes. Results of the geochemical study of quartz veins (ICP MS) and their fluid inclusions (microthermometry, gas chromatography) showed that native gold was mostly precipitated at temperatures of 230–330 °C from a low- to moderate-saline (8–12 wt.% NaCl-eq.) $H_2O$–$CO_2$–$CH_4$-bearing fluid, when weakly oxidized or near-neutral conditions, were replaced by reducing ones. No significant differences between barren milky white and ore-bearing brownish quartz veins were defined, which indicates their common formation settings and an impulse pattern of vein injection. The stable pattern of the fluid salinity, along with low hydrocarbon and $N_2$ contents, as well as a narrow range of $\delta^{18}O$ values, indicate a prevailing magmatogenic source with a certain influence of host rocks but without the influence of meteoric waters. Based on the presented data, the studied deposits were attributed to the epizonal orogenic type. This study shows the formation of lode gold deposits is possible without the participation of granite massifs.

**Keywords:** South Urals; lode gold deposits; quartz veins; geochemistry; oxygen isotopy; fluid inclusions; vapor phase; source

## 1. Introduction

Gold-quartz (lode) deposits are crucial for gold production worldwide [1–5]. These deposits are classified as orogenic and intrusion-related gold deposits [6,7]. The scale of mineralization often depends on the size of the intrusion, and the sources of ore-bearing fluids are both cooling gabbro-granite intrusions and their host volcanic-sedimentary formations [8–16]. However, determining the predominant source of fluids is often difficult due to the confusion of features, and therefore it is not easy to establish the orogenic or intrusive-related type of the deposit. In addition, the role of ultramafic-mafic intrusions in the formation of gold-quartz deposits is often underestimated, and granite massifs are usually considered the main source of heat and fluids [3–7,12,17,18].

The West Magnitogorsk Zone of the South Urals (East Bashkiria) comprises hundreds of small- and medium-sized late Paleozoic gold deposits with a different origins [19–21].

The prevalence of gabbro-granite intrusions suggests the magmatic-related genesis for most of these deposits. To elucidate the genesis of many small deposits in the West Magnitogorsk zone, we chose the Khudolaz area as one of the most typical places for the concentration of gold-quartz deposits.

The central part of the West Magnitogorsk area hosts the Khudolaz trough (Figure 1a), pierced by hundreds of small peridotite-gabbro-diorite-granite intrusions. The trough contains more than 50 small-sized gold-quartz deposits with gold reserves from several kilograms to 94 kg (>1 t in total), all poorly studied mineralogically and geochemically [20,22,23]. The exploration of these deposits started long ago, and the operation boosted to ground water (10–20 m) in the 1930s–1940s. Nowadays, these deposits are abandoned. The total production of gold from these deposits was about hundreds of kilograms, but it was poorly registered and controlled for a long time. Currently, the Khudolaz trough is of great interest due to placer gold, whose reserves are estimated at several tons. The exploration of the placers was also launched long ago, and at present, two gold mining enterprises are operating in several licensed areas at the Khudolaz trough. Notably, the famous "Irendyk bear" gold nugget with a weight of 4788 g was found right in the Khudolaz trough and has become the national treasure of Russia and Bashkiria. Recently, several nuggets weighing up to 300 g and more have also been found there. The abovementioned numerous gold-quartz deposits are considered native sources for gold placers at the Khudolaz trough.

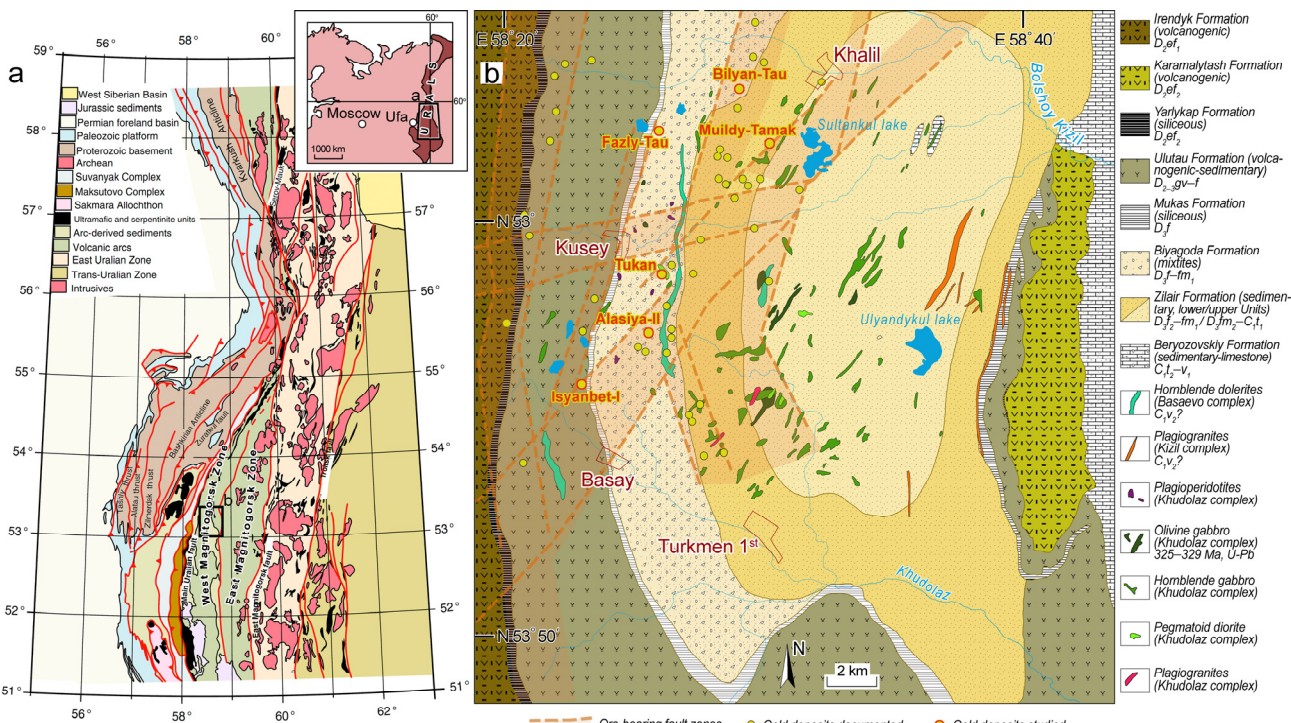

**Figure 1.** Structural-geological scheme of the South Urals after [24] (**a**) and geological map of the Khudolaz trough after [25] with changes (**b**) showing the position of intrusions of different igneous complexes and localities of small gold deposits and ore-bearing fault zones after [22].

This study provides the first findings of a detailed mineralogical study of gold-bearing quartz veins, as well as the compositional analysis and thermobaric geochemistry of fluid inclusions in quartz from six minor gold deposits at the Khudolaz trough. The paper discusses a genetic link between these deposits and intrusions from different complexes and between each other, highlights their geological settings and proposes sources for ore-bearing fluids.

## 2. Geological Setting

The Khudolaz syncline is an asymmetric steep trough composed of Devonian volcanogenic-sedimentary deposits (Figure 1b), i.e., sandstones, siltstones, siliceous rocks, minor limestones, basalts and tuffs. Syncline is about 30 km $\times$ 20 km in size, and the incline angle of the western wing is 20–30° and that of the eastern wing is 30–40°. The trough genesis is attributed to strike-slip tectonics with a pull-apart structure formed in the Middle-Late Devonian [26]. Devonian formations are broken by intrusions of the following early carbonaceous complexes: Basaevo sill-dike dolerite ($C_1$t–v), Kizil dike plagiogranite ($C_1$v), Khudolaz differentiated (U-Pb age on zircon and baddeleyite is 325–329 Ma) with numerous discordant bodies that varied in shape and size and Ulugurtau dolerite dike (Sm-Nd isochron age is $321 \pm 15$ Ma) [25]. All these intrusions are constrained by faults and fault zones of the NE strike. Nonindustrial PGE-Cu-Ni sulfide mineralization is associated with the Khudolaz complex [25].

Gold-bearing deposits and occurrences are clearly visible both on the surface and on the satellite images due to open mines, adits and ditches, which are widespread over the entire Khudolaz trough. According to [21], these deposits are constrained by several ore-bearing fault zones (Figure 1b). To study the formation settings of gold mineralization in the Khudolaz trough, we selected six deposits associated with various types of intrusions at contact with multi-compositional host rocks (Figure 1b).

## 3. Materials and Methods

Field geological works, including the description of outcrops, formation tracking and sampling, were carried out in 2022. A total of 3 to 6 samples (total of 30 samples) were collected at each deposit, including quartz veins and host rocks.

The petrographic and mineralogical study was performed using a Carl Zeiss Axioskop 40 optic microscope and a Tescan Vega Compact electron microscope (Institute of Geology of the Ural Federal Research Centre RAS, Ufa, Russia). The mineral composition was estimated using an Xplorer 15 Oxford Instruments EDS detector (Oxford Instruments plc, Oxon, UK). The following mode was set to provide measurements: 20 kV accelerated voltage, 3–4 nA earth borer current.

The elementary composition of quartz veins was assessed using an Agilent 7500cx inductively coupled plasma mass spectrometer (Agilent Technologies, Tomsk Regional Collective Use Centre, Tomsk, Russia). Samples (~1 g) were dissolved in a mixture of fluorhydric and nitric acids in a Milestone Start D microwave after preliminary curing of the mixture at a temperature of ~70 °C. Decomposition was performed at a temperature of 200 °C and at a capacity of 700 W. Next, the samples were cooled down to room temperature and washed with a 5% nitric acid solution. After that, the samples were gradually converted to chlorites and nitrates. Inner (indium solution) and outer standards (attested rock sample compositionally close to the studied sample) were applied to control changes in device sensibility. We conducted a 5 h roasting at 550 °C to determine Au. Fluorhydric acid, solutions of aqua regia and nitrohydrochloric acid (one portion of HCl to three portions of $HNO_3$) were used to prepare samples. Ten percent hydrochloric acid was applied as blank solution.

The isotope composition of oxygen as $O_2$ was defined by gas mass spectrometer FINNIGAN MAT 253 (GIN SB RAS, Ulan-Ude, Russia) using a double system of inflow in a classic variant (standard–sample). To determine $\delta^{18}O$ values, the samples were prepared by laser fluorination with the "laser ablation with oxygen extraction from silicates" mode in the presence of $BrF_5$ reagent according to the method of [27]. Only pure minerals (as fragments), with a total weight of 1.5–2.5 mg, were used for the isotope analysis of oxygen. The $\delta^{18}O$ values were determined vs. international standards NBS-28 (quartz) and NBS-30 (biotite). The accuracy of the obtained data was checked by regular measurements of an inner standard GI-1 (quartz) and a laboratory standard Polaris (quartz) of IGEM RAS. The error of the estimated $\delta^{18}O$ values was (1s) $\pm 0.2$‰.

The study of two-phase fluid inclusions in quartz was performed using a TMS-600 heating stage (Linkam) with LinkSys V-2.39 software calculating temperatures of phase transitions in the range of −196 to +600 °C and an Olympus BX-51 microscope with transmitted and reflected illumination (South Ural State University, Miass, Russia). The accuracy was ±0.1 °C in the temperature range of −20 ... +80 °C and ±1 °C outside this range. Fluid salinity was estimated on the eutectic temperatures of the solution in inclusions [28,29]. Homogenization temperatures were registered as gas bubbles vanished when the specimen was heated at this stage; the values were considered to be the minimal temperatures for mineral formation [30]. The salinity of the solutions was estimated based on the melting temperatures of the latest crystalline phases [31]. Results of the measurements were processed using Statistica software.

The bulk composition of gases in fluid inclusions was analyzed on a Tsvet-800 gas chromatograph equipped with a pyrolysis module (Institute of Geology FRS Komi SC, UB RAS, Russia). The inclusions were extracted from 0.3–0.5 g weighted portions of quartz at a temperature of 500 °C. The samples were first treated with dilute nitric acid (1:1) and washed in bidistilled water. Next, quartz grains were handpicked under a binocular microscope. The TWS-MaxiChrom software (Tavasami, Russia) was used to process the chromatographic signals. The possible relative error is 16%. Limits of detection of gas components were (mcl) $2 \times 10^{-2}$ for $N_2$ and $CO$, $3 \times 10^{-2}$ for $CH_4$ and $CO_2$, $3 \times 10^{-3}$ for $H_2O$ in the thermal conductivity detector; $CH_4$—$2 \times 10^{-9}$, $C_2H_4$—$7 \times 10^{-9}$, $C_2H_6$, $C_3H_6$—$5 \times 10^{-9}$, $C_3H_8$—$4 \times 10^{-9}$ in the flame ionization detector.

## 4. Results

### 4.1. Deposit Geology

The authors found the information about four of the deposits to be scarce in the archival geological records [32], while data on the Tukan deposit is available in the published literature [22,23]. Below is a brief description of the geological structure and host rock petrography at each deposit.

#### 4.1.1. Tukan Deposit

The deposit is located 2 km southwest of the Kusey village and spatially is associated with the gabbro intrusion of the Khudolaz complex (Figure 2a). It is one of the largest deposits with an area of 30 thousand m². Several trenches, mines and adits with depths of up to 20 m remained at the deposit area (Figure 3a). Gold-bearing quartz veins as thick as 1–10 cm produce a net of intersections both at the gabbro intrusion and in host rocks (tuff-sandstones, sandstones and silty sandstones of the Biyagoda formation) (Figure 3b). In the contact zone, the rocks are altered: gabbro and dolerites are metasomatized, which is expressed in chloritization and carbonitization, and sedimentary rocks are weakly metamorphosed, which is manifested in hornfelsing. Glide mirrors overgrown by quartz veins are observed at the eastern contact of the gabbro intrusion. They trace the fault with the dip direction of the fault displacement plane of 90–101°. The thickest veins are oriented at 174–205°, which mainly coincides with the orientation of dolerite dikes at the Ulugurtau complex cutting the gabbro intrusion. Segregates of native gold up to several millimeters in size are located in cracks of quartz, carbonate-quartz and quartz-dolerite veins, as well as in metasomatic dolerite near the contact with quartz. According to [32], the average Au content in ores of the Tukan deposit is 5.2 ppm and, in certain samples, reaches up to 100 ppm.

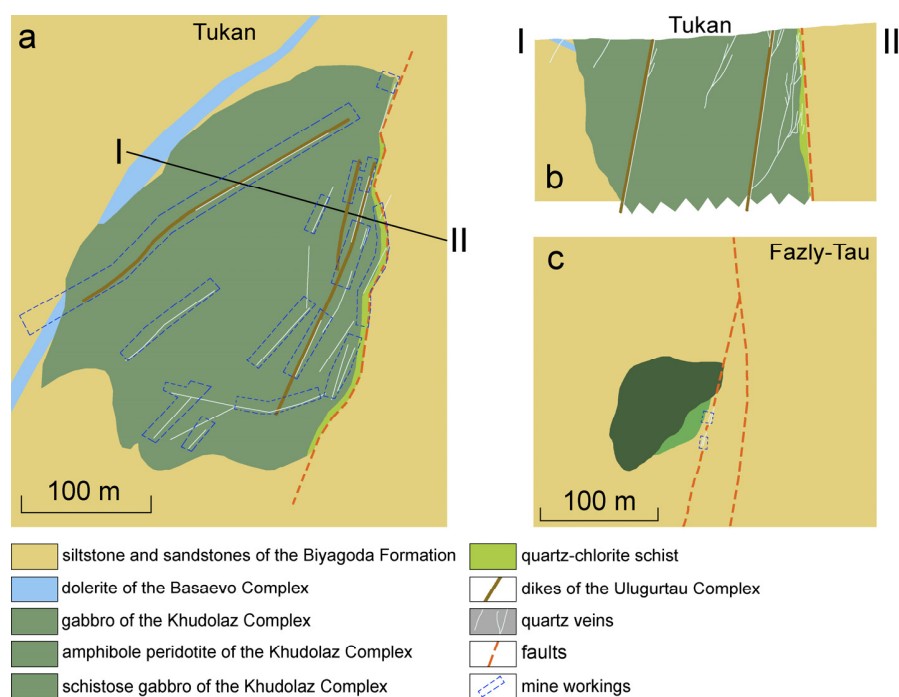

**Figure 2.** Schematic geological maps of Tukan (**a**) after [23] and Fazly-Tau (**c**) [this study]. Gold deposits of the Khudolaz area: (**a**) map in plan, (**b**) geological section across I–II line.

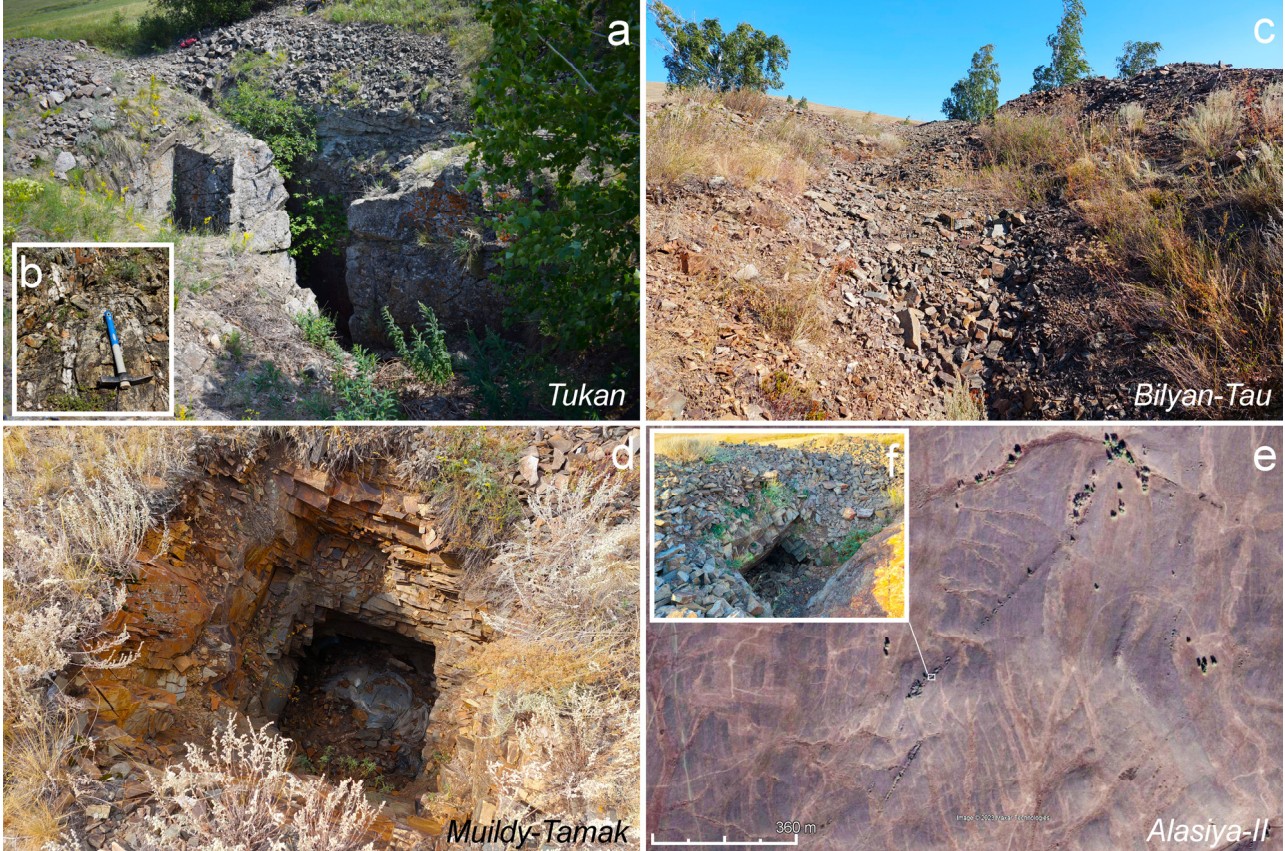

**Figure 3.** Ancient open-pits at some small gold deposits in the Khudolaz area: (**a**) pits of the Tukan deposit, (**b**) outcrop of gold-bearing quartz veins cutting host gabbro in the Tukan deposit, (**c**) dumps on the Bilyan-Tau deposit, (**d**) pit on the Muildy-Tamak deposit, (**e**) trenches of the Alasiya-II deposit seen via satellite image and (**f**) inclined pit in the Alasiya-II deposit.

### 4.1.2. Bilyan-Tau Deposit

The deposit is located 3 km west of the Khalil village (Figure 1b). It is confined to contacts of a dolerite sill at the Basaevo complex and a small gabbro intrusion of the Khudolaz complex with silty sandstones, tuffs and basalts of the Biyagoda formation. Judging by the compositing debris in dumps, dikes of the Ulugurtau complex widely occur at the deposit area. It is the largest deposit with an area of 66 thousand m², hosting an abandoned mine shaft, numerous ditches and bell pits. Abundant dumps indicate intense mining of the deposit (Figure 3c). Most trenches have the NE strike (~30°). In the central part of the area, there is a zone of intense faulting with a thickness of up to 60 m, a dip azimuth of 285° and a dip angle of 55–60° [20]. An intrusion of altered taxitic gabbro composed of plagioclase, hornblende and clinopyroxene is elongated along an azimuth of 340°, while a hornblende dolerite sill with an unclear thickness has a meridional strike. From west to east, tuff sandstones are replaced by basalt tuffs and rare basalts, containing siltstone xenoliths up to 10 cm in size. Native gold is located in quartz veins 10–15 cm thick (and possibly more), producing a dense net of intersections in dolerites, gabbro, basalt tuffs and siltstones (Figure 4a,b). In one of the test pits, the siltstone bed position was measured to have a strike azimuth of 15° and an incidence angle of 23°. We studied only ore samples from dumps because of poor exposure to the ores. According to [32], the average Au content in ores was 11.9 ppm.

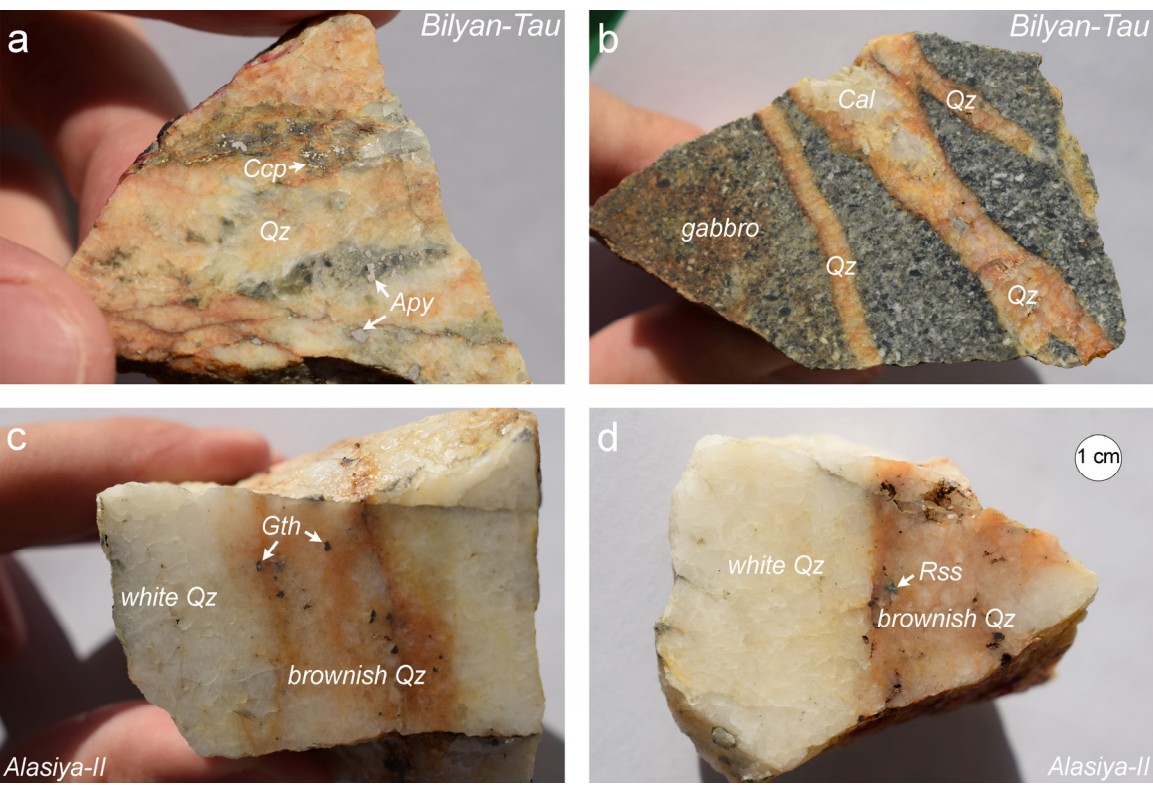

**Figure 4.** Hand specimens of quartz veins samples from some gold deposits in the Khudolaz area. Apy = arsenopyrite, Cal = calcite, Ccp-chalcopyrite, Gth = goethite, Qz = quartz, Rss = rosasite. (**a**) sulfide and sulfoarsenide mineralization in brownish quartz vein, (**b**) carbonate-quartz veins cutting host gabbro, (**c**) and (**d**) zoning white-brownish quartz veins.

### 4.1.3. Fazly-Tau Deposit

The deposit is 4 km southwest of the Bilyan-Tau deposit (Figure 1b). There are no data on it, even in archive geological records. Our research indicates that the deposit is confined to the eastern contact of the ultrabasic intrusion of the Khudolaz complex with polymictic sandstones of the Biyagoda formation (Figure 2c). In the endocontact of the intrusion, schistose gabbro is observed, and the host sandstones are metasomatized to the appearance

of chlorite and albite. The area of ground with several trenches and pits is not more than 1000 m$^2$. In the western wall of one pit, we tested brownish quartz veins with an eastern strike of 295° and an incidence angle of 41°. The thickness of quartz veins is not more than 10 cm, and they produce a dense net of intersections in schistose gabbro. The intrusion occurs as a stock of hornblende peridotites with disseminated sulfide mineralization. A sandstone suite comprising it has a dip azimuth of 268° and an incidence angle of 40°.

### 4.1.4. Muildy-Tamak Deposit

The deposit is southwest of the Khalil village and west of Lake Sultankul (Figure 1b). The deposit is confined to the contact of a dike of the Ulugurtau complex with siltstones of the Zilair formation. The thickness of the dike composed of carbonatized hornblende dolerites is unclear due to its poor exposure. On the contact site of the dike, siltstones are hornfelsed, the dip azimuth of the siltstone sequence is 70°, and the incidence angle is 15°. One 300 m long ditch and several partly collapsed mines remain in the area (Figure 3d). Native gold is located in a major quartz vein with a strike of 15°; it is also probably found in fine veins branching off from the major vein. According to [32], the average Au content in rocks from these dumps is 0.2 ppm.

### 4.1.5. Alasiya-II Deposit

The deposit is located 2 km south of the Tukan deposit (Figure 1b). It is confined to the contact of olivine gabbro intrusion of the Khudolaz complex and hornfelsed polymictic sandstones of the Biyagoda formation. Two ditches (Figure 3e) with inclined mines of western strike remained in a 900 m long area (Figure 3f). Native gold is located in major quartz veins with strikes of 36° and 50°; the average gold content in the ore was 6.3 ppm as per [32]. We sampled a zonal (brownish-white) quartz vein (Figure 4c,d) from the eastern wall of the trench.

### 4.1.6. Isyanbet-I Deposit

The deposit occurs 2.5 km north of the Basay village (Figure 1b) in the carbonatized polymictic sandstones of the Mukas (?) formation. The area of the deposit is about 3 thousand m$^2$; it hosts abandoned and partly collapsed ditches and pits that are about 3 × 3 and 4 × 4 m in diameter. Fragments of hornblende gabbro and dolerites were found in the dumps. Native gold occurs in quartz veins, producing a net of intersections in sandstones, and some veins form intergrowths with dolerite. The largest veins with a thickness of up to 10 cm have a NE strike of about 30–40°. According to [32], the average Au content in the ore was 21.1 ppm. We tested white and brownish veins from dumps because studying the outcrops on the mine walls was too dangerous.

### 4.2. Mineralogy of Gold-Bearing Quartz Veins

Mineralogical studies showed that the Khudolaz area gold deposits differed both in prevalent ore minerals in ore-bearing quartz veins and in mineralogical diversity in general (Figure 5). Quartz veins of the Tukan and Fazly-Tau deposits contain an insignificant amount of ore minerals, while veins of the Bilyan-Tau and Alasiya-II deposits are characterized by richer mineral diversity. Quartz veins of the Muildy-Tamak and Isyanbet-I deposits contain moderate amounts of ore minerals. We defined two types of quartz veins in each deposit, i.e., (1) early milk-white barren (or weakly ore-bearing) and (2) brownish ore-bearing. The color of brownish quartz is caused by the presence of fine blades of iron hydroxides on the surface of the cracks. This particular quartz contains dissemination of not only native gold but also other minerals such as oxides, hydroxides, and sulfides. The content of sulfides and sulfoarsenides in quartz veins varies from traces (Tukan deposit) to 7 wt % (Bilyan-Tau deposit). The two types of quartz occur both as individual veins and veinlets and as associated zonal veins, where white quartz is observed at the rims, and brownish quartz forms the central part of the vein (Figure 4c,d). Mottled and banded brownish-white veins are quite common (Figure 4a). Native gold was studied only at the

Tukan deposit, where it is located in cracks and rarely intergrows with pyrite (Figure 6a). The studied gold grains are up to 1.5 mm in size with a fineness of $871.0 \pm 8.3$‰ [23]. An insignificant amount of native gold with a fineness of 834‰–853‰ was found as thin veinlets in hessites within a quartz-carbonate vein, where gold is associated with low-temperature Pb-Zn-Te phases.

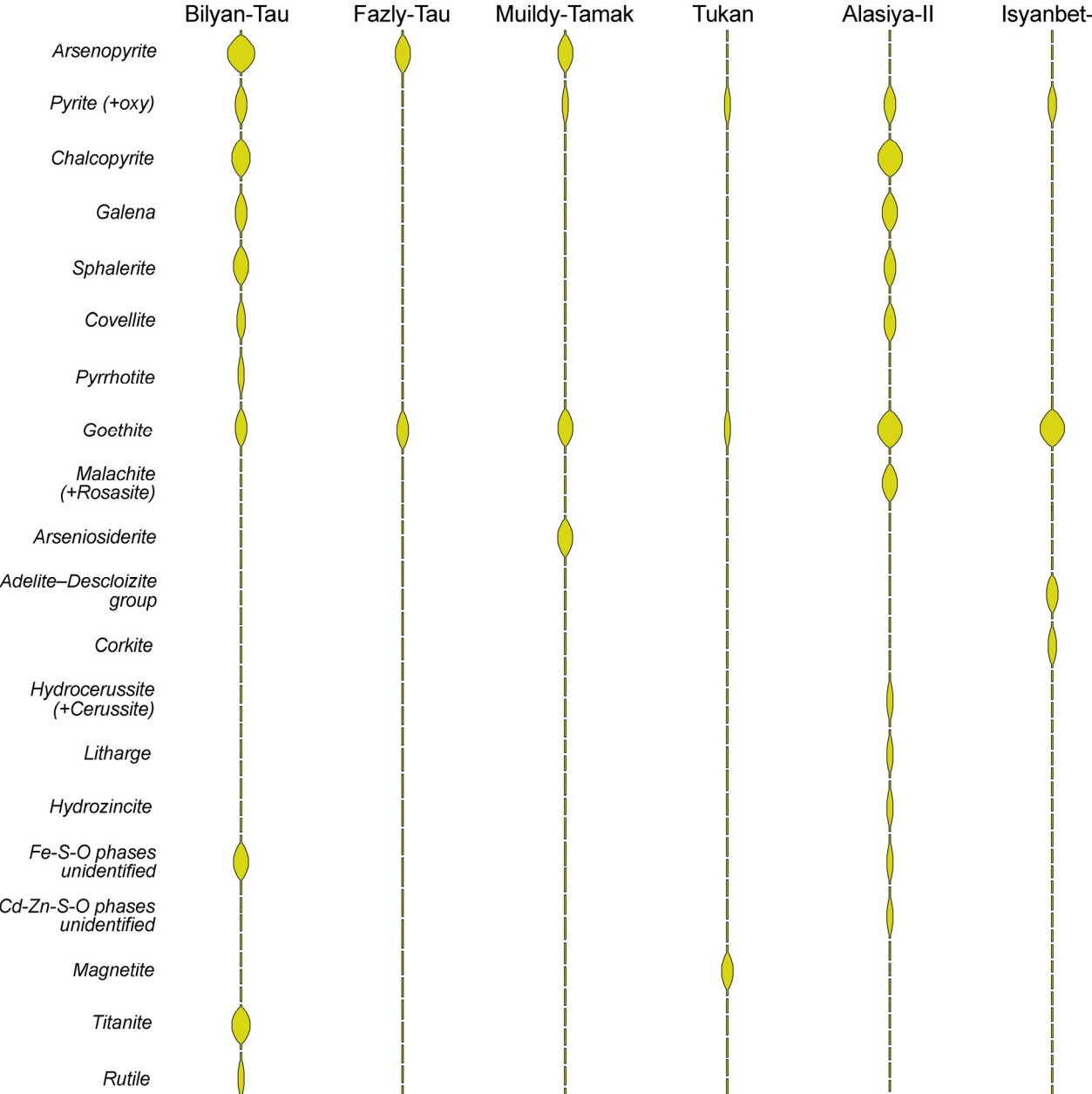

**Figure 5.** Relative content of ore minerals (except native gold) in gold-bearing quartz veins from gold deposits in the Khudolaz area.

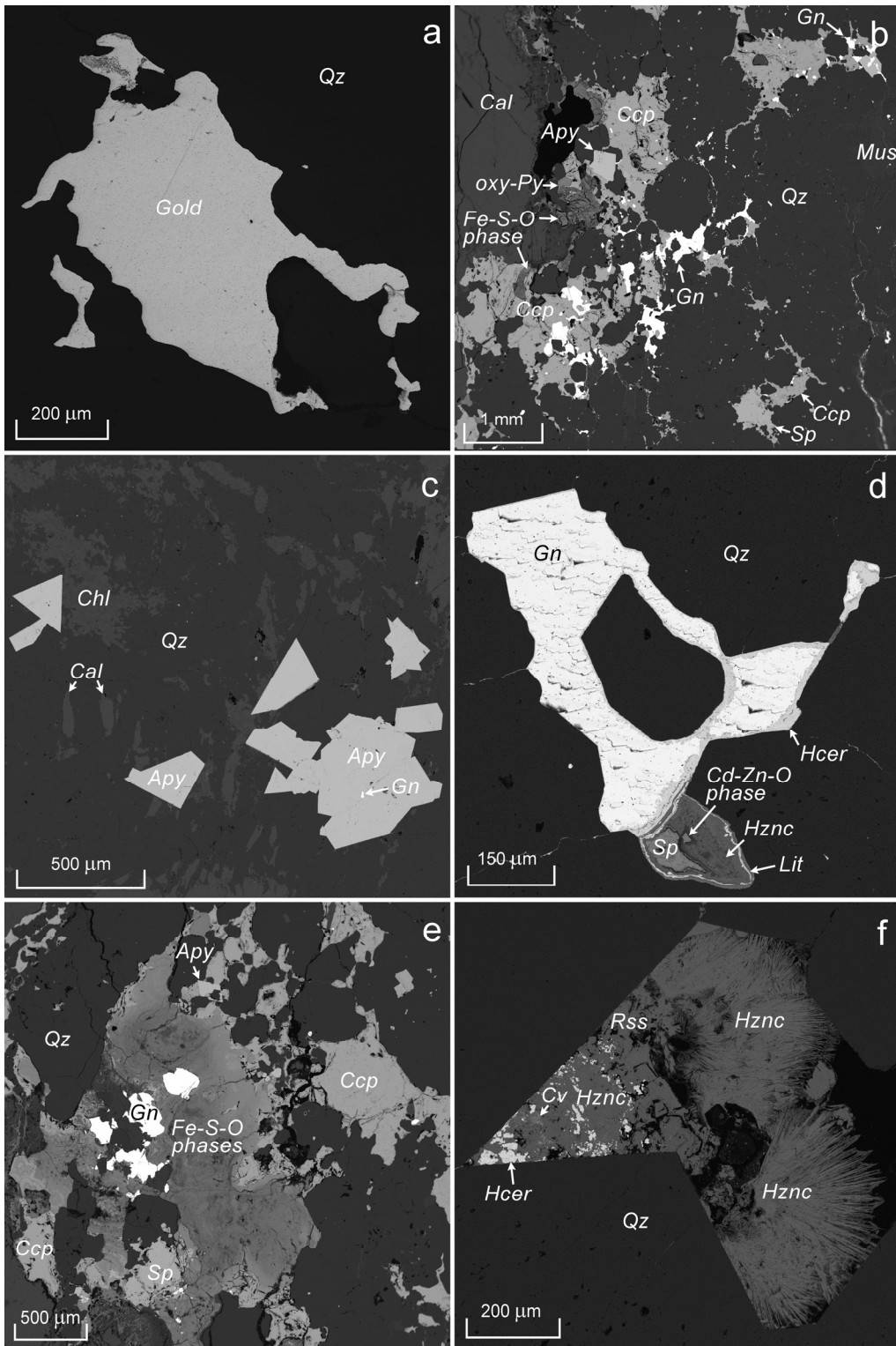

**Figure 6.** Mineralogy of gold-bearing quartz veins from small gold deposits of the Khudolaz area (BSE images): (**a**) xenomorphic gold grain between quartz grains (Tukan deposit), (**b**) polymineralic sulfide assemblage (Bilyan-Tau deposit), (**c**) euhedral arsenopyrite grains (Bilyan-Tau deposit), (**d**) large grain of galena with secondary minerals replaced sphalerite (Alasiya-II deposit), (**e**) unidentified Fe-S-O phases replaced minerals of a polymineralic sulfide assemblage (Bilyan-Tau deposit), (**f**) secondary minerals replaced primary sulfides in space between quartz grains (Alasiya-II deposit). Apy = arsenopyrite, Cal = calcite, Ccp = chalcopyrite, Chl = chlorite, Cv = covellite, Hcer = hydrocerussite, Hznc = hydrozincite, Gn = galena, Lit = litharge, Mus = muscovite, Py = pyrite, Qz = quartz, Rss = rosasite, Sp = sphalerite.

Goethite is a ubiquitous ore mineral of quartz veins that substituted earlier oxides and sulfides. The specific feature of goethite in all deposits is its enrichment by Cu (up to 10.2%), Zn (up to 2.7%), Pb (up to 8.0%) and occasionally Ca (up to 2.8%) and P (up to 0.3%). At places, products of sulfide replacement appear as a finely dispersed mixture of goethite with other aqueous minerals, i.e., hydrocerussite, hydrozincite, rosasite, etc. In all deposits, except for Fazly-Tau, quartz veins contain a minor amount of pyrite that produces scattered dissemination or small (less than 50 µm) inclusions in other sulfides. However, the oxygen-bearing phase is the most common (Figure 6b). It is likely to occur as an oxidized pyrite, where some sulfur is replaced by oxygen. In half of the studied deposits, one of the leading ore minerals is arsenopyrite, commonly represented by large (up to 500 µm) idiomorphic crystals (Figure 6b,c). Other minerals are observed only in one or two deposits.

Chalcopyrite, covellite, sphalerite and galenites are often found as joint xenomorphic polymineral aggregates in quartz veins of the Bilyan-Tau and Alasiya-II deposits (Figure 6b,e). Chalcopyrite and galena also occur as veinlets or individual grains with sizes up to 1–2 mm (Figure 6d). In addition, with a total of 98 to 100% in analysis, unidentified Fe-S-O phases with admixtures of Cu, Pb, Zn, As and Si were found in the quartz veins of these two deposits. The stoichiometric formula for most of them was calculated based on eight atoms: $(Fe_{1.60-2.07}, Cu_{0.02-0.17}, Pb_{0.03-0.08}, Zn_{0.01-0.03})_{1.88-2.14} (S_{0.83-0.98}, Si_{0.03-0.24}, As_{0-0.12})_{0.98-1.12} O_{4.88-5.04}$. Such an oxysulfide ($Fe_2SO_5$), as defined by [33], was also discovered in ores and dumps of Upper Silesia (Poland). It is reported to be a crystalline matter related to the orthorhombic system, according to powder diffractometry data.

Quartz veins of the Alasiya-II deposit often contain aggregates of rosasite and malachite, both as individual radial segregations and as intergrowths with other low-temperature phases that replaced primary sulfides (Figure 6f). Various unidentified Cd-Zn-O and Cd-S-Zn-O phases with an analysis sum of 88 to 100% with admixtures of Pb, Ca, Cu and Si are widespread here as well. These phases produce collomorphic aggregates intergrown with hydrozinc after sphalerite (Figure 6d). The following formula was derived (for eight atoms) for an analysis of the Cd-Zn-O phase with a sum of 98.73 wt.%: $(Zn_{0.59}, Ca_{0.25}, Fe_{0.13}, Pb_{0.04})_{1.00} Cd_{1.09} O_{5.91}$. The following formula was derived (for 14 atoms) for an analysis of the Cd-S-Zn-O phase with a sum of 100%: $(Fe_{1.38}, Zn_{0.70}, Pb_{0.05})_{2.09} Cd_{2.79} (S_{0.77}, Si_{0.29})_{3.06} O_{6.06}$. There are no such compounds in mineralogical databases yet.

Found in minor amounts jointly with goethite and other low-temperature secondary phases are arseniosiderite (Muildy-Tamak); mottramite and corkite (Isyanbet-I); and hydrocerussite, litharge and hydrozincite (Alasiya-II). All these minerals commonly produce veinlets with fine xenomorphic segregations along the rims or fractures of primary sulfides. Quartz veins of the Bilyan-Tau deposits contain titanite crystals up to 0.3 mm long and small segregations of rutile associated with chlorite and muscovite. In addition, quartz veins of the Bilyan-Tau deposit contain abundant calcite and minor gypsum associated with goethite. Magnetite often occurs in quartz veins of the Tukan deposit. Calcite (Figure 4b), chlorite, and muscovite occur as late veinlets in quartz veins in all deposits.

### 4.3. Geochemistry of Quartz Veins

The trace element composition of quartz from ore-bearing and barren veins was studied in seven samples, two of them were from the Tukan deposit, and these was one of each from the other deposits. For our analyses, we tried to select areas of quartz veins that were visibly less covered by various inclusions. The results are provided in Tables 1 and 2.

**Table 1.** Concentrations of chalcogen, ferrous and nonferrous metals in quartz veins from gold deposits of the Khudolaz area (ppm).

| # | 1 | 2 | 3 | 4 | 5 | 6 | 7 |
|---|---|---|---|---|---|---|---|
| | D$_2$-36a | D$_2$-36b | K$_{22}$-20c | K$_{22}$-23 | K$_{22}$-29a | K$_{22}$-32b | K$_{22}$-38a |
| Element | Tukan | | Muildy-Tamak | Isyanbet-1 | Alasiya-II | Fazly-Tau | Bilyan-Tau |
| Ti | 26 | 286 | 23 | 40 | 38 | 229 | 165 |
| Cu | 3.6 | 2.6 | 15.1 | 7.6 | 432 | 20 | 13.7 |
| Zn | 2.4 | 18 | 2.6 | 2.3 | 307 | 13.9 | 7.2 |
| Pb | 2.3 | 2.9 | 5 | 0.38 | 1805 | 15.7 | 10.2 |
| Fe | 1329 | 3344 | 3369 | 39 | 3339 | 4643 | 4067 |
| Se | 0.032 | 0.410 | 0.580 | 0.530 | 1.200 | 0.590 | 0.084 |
| Ag | 0.007 | 0.072 | 0.160 | 2.400 | 3.700 | 0.006 | 0.280 |
| Cd | 0.067 | 0.180 | 0.070 | 0.003 | 19.000 | 0.130 | 0.067 |
| Sb | 0.620 | 1.200 | 3.000 | 0.700 | 1.500 | 0.340 | 0.620 |
| Te | 0.031 | 0.045 | 0.053 | 0.018 | 4.700 | 0.030 | 0.590 |
| Au | 0.059 | 0.450 | 0.980 | 0.480 | 0.066 | 0.042 | 0.110 |
| Bi | 0.018 | 0.420 | 0.180 | 0.008 | 0.570 | 0.080 | 0.810 |

Note: 1—barren quartz, and 2–7—ore-bearing quartz.

**Table 2.** Concentrations of REEs in quartz veins from gold deposits in the Khudolaz area (ppm).

| # | 1 | 2 | 3 | 4 | 5 | 6 | 7 |
|---|---|---|---|---|---|---|---|
| | D$_2$-36a | D$_2$-36b | K$_{22}$-20c | K$_{22}$-23 | K$_{22}$-29a | K$_{22}$-32b | K$_{22}$-38a |
| Element | Tukan | | Muildy-Tamak | Isyanbet-1 | Alasiya-II | Fazly-Tau | Bilyan-Tau |
| La | 1.700 | 1.500 | 0.190 | 0.103 | 0.105 | 0.560 | 0.410 |
| Ce | 3.200 | 3.300 | 0.440 | 0.190 | 0.220 | 1.190 | 0.840 |
| Pr | 0.410 | 0.510 | 0.059 | 0.022 | 0.035 | 0.150 | 0.103 |
| Nd | 1.800 | 2.400 | 0.240 | 0.087 | 0.160 | 0.610 | 0.430 |
| Sm | 0.530 | 0.840 | 0.058 | 0.022 | 0.041 | 0.130 | 0.099 |
| Eu | 2.000 | 3.000 | 0.016 | 0.009 | 0.048 | 0.048 | 0.032 |
| Gd | 0.530 | 0.780 | 0.056 | 0.019 | 0.042 | 0.130 | 0.092 |
| Tb | 0.097 | 0.140 | 0.009 | 0.003 | 0.007 | 0.022 | 0.016 |
| Dy | 0.490 | 0.700 | 0.043 | 0.026 | 0.044 | 0.130 | 0.098 |
| Ho | 0.099 | 0.140 | 0.008 | 0.005 | 0.009 | 0.030 | 0.019 |
| Er | 0.260 | 0.340 | 0.026 | 0.016 | 0.026 | 0.075 | 0.050 |
| Tm | 0.031 | 0.036 | 0.002 | <0.001 | 0.002 | 0.015 | 0.009 |
| Yb | 0.270 | 0.320 | 0.028 | 0.022 | 0.032 | 0.087 | 0.057 |
| Lu | 0.029 | 0.035 | 0.004 | 0.003 | 0.005 | 0.013 | 0.009 |

Note: 1—barren quartz, and 2–7—ore-bearing quartz.

Concentrations of admixture elements in quartz veins from different deposits vary greatly, which may be caused by a varying degree of enrichment with accessory phases (ore minerals, calcite, etc.), despite their visual "purity". Figure 7 provides a histogram of the composition of 12 elements in quartz veins. Quartz veins at the Alasiya-II deposit (sample K$_{22}$-29a) are the richest in Cu, Zn, Pb, Se, Ag, Cd and Te. The highest Au content was detected in quartz veins of Muildy-Tamak (0.98 ppm) (sample K$_{22}$-20c), Isyanbet-I (0.48 ppm) (sample K$_{22}$-23) and Tukan (0.45 ppm) (sample D$_2$-36b); at the other deposits, it did not increase past 0.1 ppm. Analysis of element correlations for seven samples indicated a significant correlation of gold with antimony only (correlation coefficient +0.80), while gold only scarcely correlated with the other elements. The difference between concentrations of chalcogen, ferrous and nonferrous metals (up to 10 times) between a barren and an ore-bearing quartz from the Tukan deposit (sample D$_2$-36a and D$_2$-36b, Table 1) is fairly representative, while concentrations of rare earth elements (REE) in them are similar (sample D$_2$-36a and D$_2$-36b, Table 2).

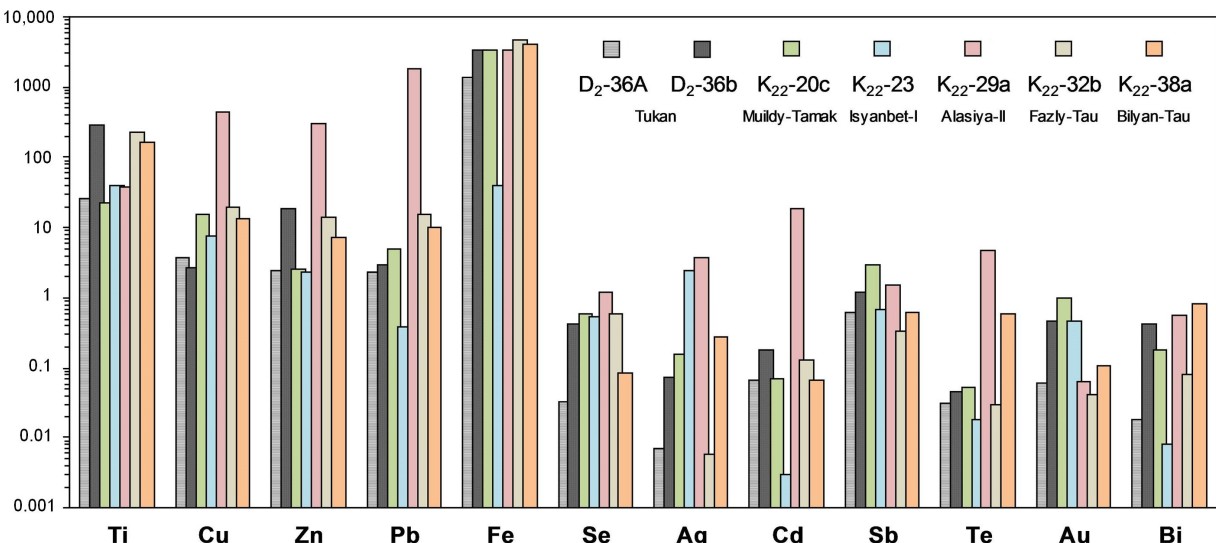

**Figure 7.** Histogram of chalcogen, ferrous and nonferrous metal concentrations in quartz veins from gold deposits in the Khudolaz area.

Quartz veins of the Tukan deposit are marked by the highest REE concentrations with $\sum$REE = 11.4–14.0 ppm, while at other deposits, $\sum$REE varies from 0.8 to 3.2 ppm. This may be caused by the fact that the analyzed quartz veins of the Tukan deposit contain calcite inclusions. Distribution spectra show a slight incline (Figure 8), and the $(La/Yb)_N$ ratio ranges from 2.2 to 4.9. A strong positive Eu-anomaly was revealed in both samples of the Tukan deposit (Eu/Eu* = 11.5 and 11.3) and in one sample from the Alasiya-II deposit (Eu/Eu* = 3.5). Other deposits show a slight negative (Muildy-Tamak) or positive Eu-anomaly (Bilyan-Tau, Fazly-Tau and Isyanbet-I). In addition, a significant negative Tm-anomaly (Tm/Tm* = 0.34–0.50) was revealed in quartz veins of the Muildy-Tamak, Alasiya-II and Isyanbet-I deposits.

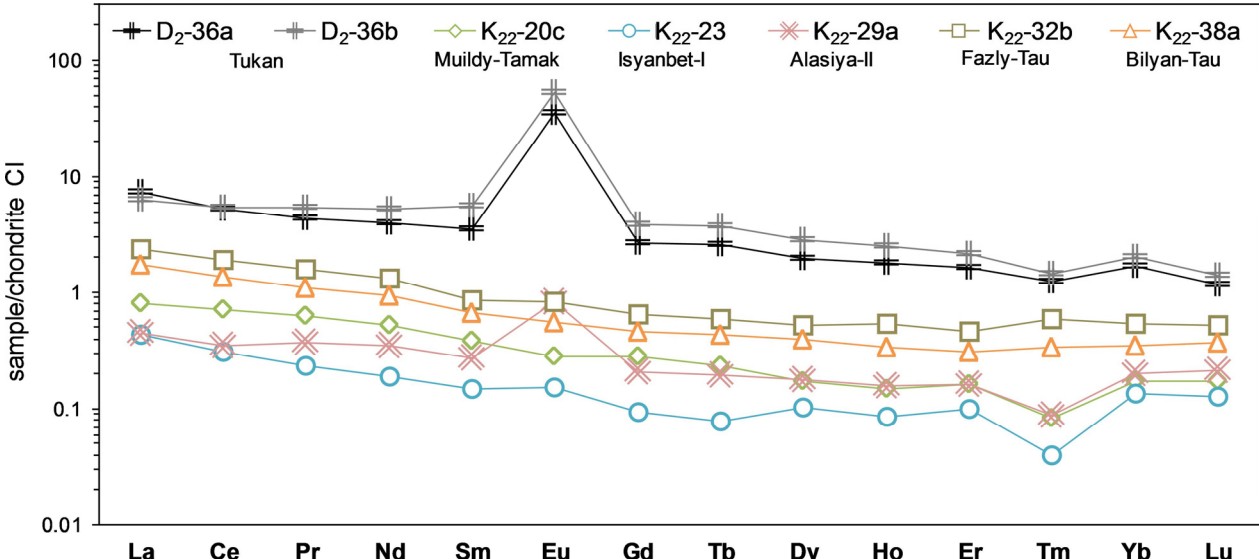

**Figure 8.** Chondrite-normalized REE patterns of quartz veins from gold deposits in the Khudolaz area.

### 4.4. Fluid Inclusion Data

To estimate the temperatures of mineral formations, salt composition and salinity of ore-forming fluids and fluid inclusions in polished thin sections of quartz were analyzed. Samples from ore-bearing brownish quartz veins and barren white veins were used as study

material. Two-phase (vapor–liquid, VL) fluid inclusions in quartz were analyzed. Based on optical observations and criteria of E. Roedder [30] recognized primary and primary-secondary fluid inclusions that occur as single inclusions and few clusters in central parts of grains 15 μm in size, as well as 5 μm secondary inclusions that mark fractures passing through several quartz grains (Figure 9). In addition, two-phase inclusions in quartz are associated with rounded mainly single-phase fluid and gas inclusions up to 5 μm in size. Table 3 and Figure 10 provides the results of fluid inclusion thermometry.

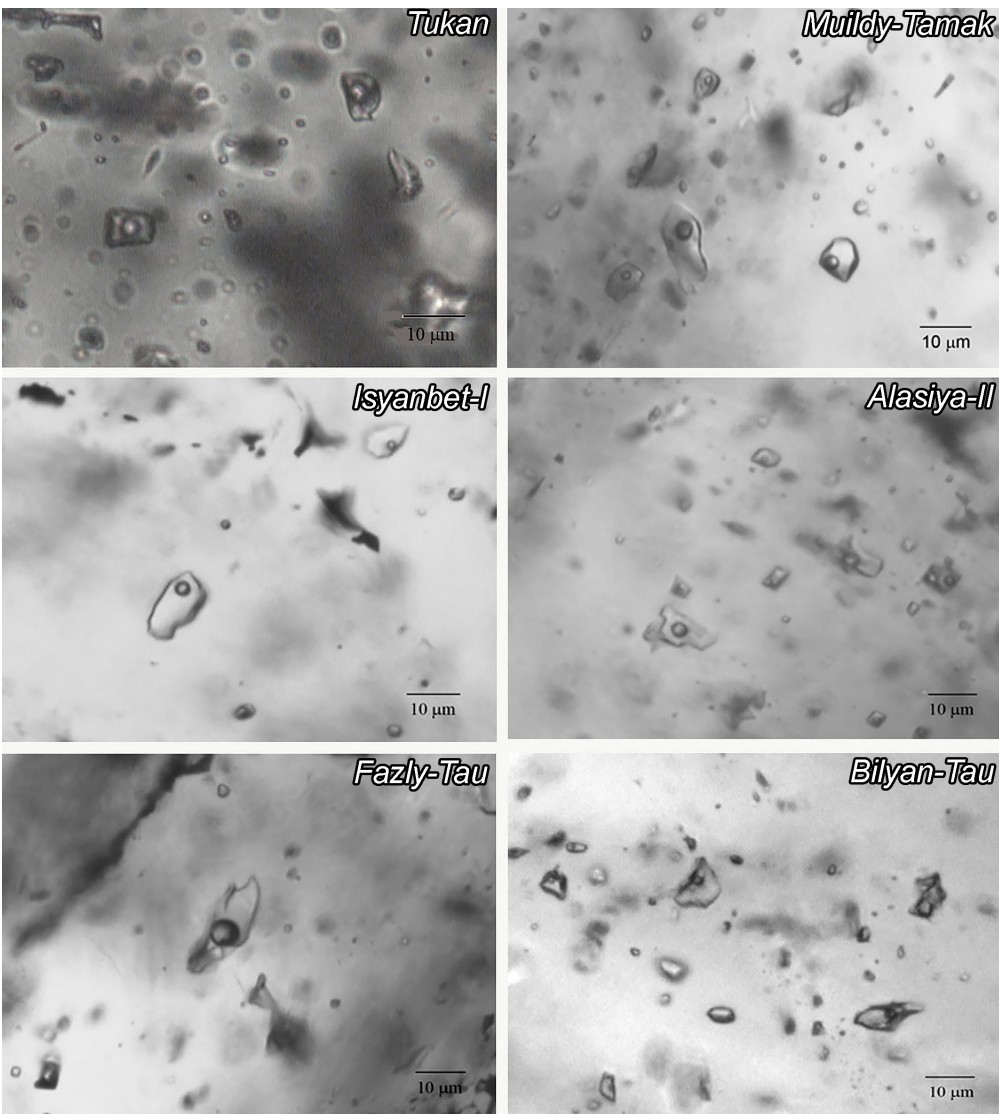

**Figure 9.** Fluid inclusions in quartz veins from gold deposits in the Khudolaz area.

At the *Tukan* deposit, fluid inclusions in both quartz varieties from the single zonal vein are 10–12 μm and are isometric or oval in shape. Gas vacuoles occupied 15%–20% of the inclusion volume. The analyzed inclusions contained solutions with eutectic temperatures of −21–−22 °C, indicating the presence of Na chlorides in the fluid. Homogenization temperatures of fluid inclusions in milk-white quartz (sample 731A-b) are higher and vary in a narrower range (230–254 °C, $n = 46$) compared to inclusions in brownish quartz (sample 731A-o) (186–230 °C, $n = 20$). The salinity of the solutions varies from 5 to 9 wt.% NaCl-eq. in inclusions in white quartz and from 4 to 8 wt.% NaCl-eq. in solutions in brownish quartz. More data have been formerly provided in [23].

**Table 3.** Results of the fluid inclusion study in quartz veins from gold deposits in the Khudolaz area.

| Sample | The Association of FI (Type) | N | $T_{hom}$, °C | $T_{eutectic}$, °C (Main Fluid's Salt) | $T_{ice\ melting}$, °C | C, wt.% NaCl-eq. |
|---|---|---|---|---|---|---|
| | | | Tukan | | | |
| 731A-b | P, PS (VL) | 46 | 230–254 | −21 … −22 (NaCl) | −3.2 … −6.3 | 5.3–9.2 |
| 731A-o | | 20 | 185–230 | | −2.5 … −4 | 4.2–6.4 |
| | | | Muildy-Tamak | | | |
| $K_{22}$-20b | P, PS (VL) | 35 | 280–318 | −23 … −24 (NaCl ± KCl) | −2.8 … −4.6 | 4.6–7.3 |
| $K_{22}$-20c | | 35 | 225–255 | −23 … −28 (NaCl ± KCl) | −5 … −6.8 | 7.9–10.2 |
| | | | Isyanbet-I | | | |
| $K_{22}$-23a | P, PS (VL) | 30 | 294–324 | −21 … −23 (NaCl ± KCl) | −5.2 … −7 | 8.1–10.5 |
| $K_{22}$-23b | | 35 | 306–337 | −23 … −25 (NaCl ± KCl) | −6.5 … −8.3 | 9.9–12.0 |
| | | | Alasiya-II | | | |
| $K_{22}$-29a | P, PS (VL) | 35 | 320–354 | −22 … −24 (NaCl ± KCl) | −4 … −6 | 6.4–9.2 |
| | | | Fazly-Tau | | | |
| $K_{22}$-32a | P, PS (VL) | 32 | 230–254 | −22 … −24 (NaCl ± KCl) | −3.9 … −6 | 5.9–9.3 |
| $K_{22}$-32b | | 20 | 220–238 | −21 … −22 (NaCl) | −3.6 … −6.1 | 5.7–8.9 |
| | | | Bilyan-Tau | | | |
| $K_{22}$-38a | P, PS (VL) | 35 | 194–285 | −21 … −23 (NaCl ± KCl) | −3.3 … −7 | 5.4–10.5 |
| $K_{22}$-38b | | 37 | 164–206 | −23 … −28 (NaCl ± KCl) | −3.2 … −5.7 | 5.3–8.8 |

Note: Inclusions: P—primary, PS—pseudosecondary, VL—two-phase (vapor–liquid); N—number of measurements, $T_{hom}$—homogenization temperature, $T_{eutectic}$—eutectic temperature, $T_{ice\ melting}$—temperature of melting of the latest ice crystal, C—salinity.

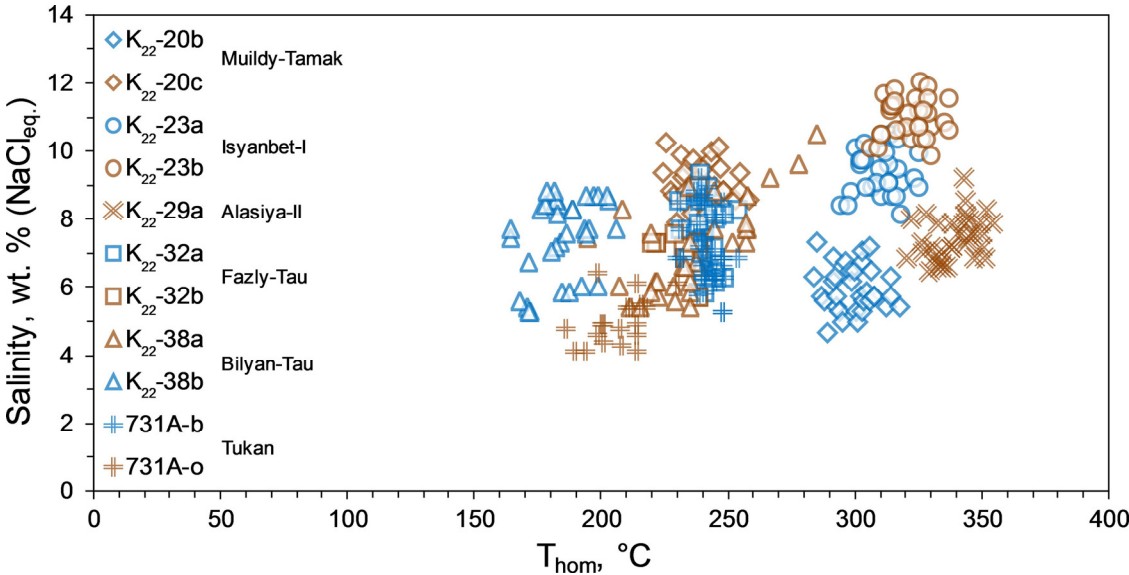

**Figure 10.** Homogenization temperature ($T_{hom}$) vs. salinity plot for fluid inclusions in quartz veins from gold deposits in the Khudolaz area. Brownish symbols—ore-bearing quartz. Blue symbols—barren quartz.

At the *Muildy-Tamak* deposit, two-phase fluid inclusions were studied in large semi-transparent grains of brownish-white (sample $K_{22}$-20c) and milk-white (sample $K_{22}$-20b)

quartz from major veins. Fluid inclusions have a size of up to 20 μm, are isometric and elongated in shape with crystallographic elements. They are either isolated or occur as small groups of 2–3 inclusions. Gas bubbles are distinct and occupy up to 15%–20% of the inclusion volume. Aqueous chlorides of Na and K were detected in the salt composition at the liquid phase of fluid inclusions. Eutectic temperatures range from −23 . . . −24 to −23 . . . −28 °C, respectively (*n* = 70). In the liquid phase, inclusions were homogenized at 280–318 °C in brownish-white quartz (mode 300–305 °C) and at 225–255 °C in white barren quartz (mode 240–245 °C). The salinity is higher in inclusions in white quartz (7.9–10.2 wt.%, mode 8.5–9 wt.%), and it varies from 4.6 to 7.3 wt.% NaCl-eq. in brownish-white quartz (mode 5.5–6).

At the *Isyanbet-I* deposit, massive milk-white quartz (sample $K_{22}$-23a) and fractured brownish quartz (sample $K_{22}$-23b) were studied. Two-phase fluid inclusions 10–15 μm in size were observed in transparent or semi-transparent quartz grains. The inclusions have a round isometric shape and occur separately, all confined to the margins of quartz grains. Eutectic temperatures vary from −23 to 25 °C (*n* = 35) and characterize the salt system NaCl-KCl-$H_2O$. The homogenization temperatures range from 306 to 337 °C (*n* = 35). The fluid salinity in the inclusions was 9.9–12.0 wt.% NaCl-eq. Fluid inclusions in white barren quartz are marked by eutectic temperatures of −21 . . . −23 °C (*n* = 30) and the aqueous NaCl-KCl composition of the fluid. In the liquid phase, the inclusions homogenized at temperatures of 294–324 °C (mode 310–315 °C). The fluid salinity was 8.1–10.5 wt.% NaCl-eq.

Two-phase fluid inclusions in mottled brownish-white ore-bearing quartz (sample $K_{22}$-29a) from a thick vein from the *Alasiya-II* with a fluid eutectic temperature of −22 to −24 °C characterize the water–salt system NaCl—KCl-$H_2O$ (*n* = 35). In the liquid phase, the inclusions homogenized at temperatures of 320 to 354 °C (*n* = 35). Fluid salinity values vary from 6.4 to 9.2 wt.% NaCl-eq.

At the *Fazly-Tau* deposit, white quartz with chlorite inclusions (sample $K_{22}$-32a) and brownish-white quartz with sulfide inclusions (sample $K_{22}$-32b) were studied. The eutectic temperature of the inclusions ranged from −21 to −24 °C, which indicates the presence of NaCl and KCl in aqueous fluid. The inclusions are characterized by melting temperatures of the latest crystalline phases of −3.6 to −6.1 °C, and the salinity of the trapped fluid was 5.7–9.3 wt.% NaCl-eq. (mode 7.0–7.5 wt.%) In the liquid phase, the inclusions were homogenized at temperatures of 220 to 254 °C.

At the *Bilyan-Tau* deposit, mottled brownish-white ore-bearing quartz (sample $K_{22}$-38a) from a large vein and white barren quartz from thin veinlets (sample $K_{22}$-38b) were studied. Fluid inclusions show ice melting temperatures of −3.2 to −7.0 °C. Hence, their fluid salinity was 5.3–10.0 wt.% NaCl-eq. In the liquid phase, the inclusions in brownish quartz were homogenized at temperatures of 194–285 °C. Inclusions in white quartz were homogenized at lower temperatures of 164–206 °C.

### 4.5. Gas-Phase Analysis

The results of gas chromatography (Table 4, Figure 11) showed that $CO_2$ was the main volatile component of fluid inclusions in quartz in all deposits (11.3–52.2 ppm); the total amount of methane and heavier hydrocarbon gases (HG) was not higher than 7 ppm. The amount of $H_2O$ in quartz is 332–1780 ppm, while $H_2$ and CO are below the detection limit. The amount of $N_2$ in fluids is low and close to or below the detection limit (≤0.1 ppm); only inclusions from quartz of the Tukan deposit contain 0.19 ppm $N_2$. The amount of $CH_4$ (0.7–5.9 ppm) correlates with the $H_2O$ content (correlation coefficient is 0.787), while $CO_2$ shows almost no correlation with the $H_2O$ content. Fluid inclusions in the Tukan deposit are the most $CH_4$-depleted (0.7 ppm). The most $H_2O$-enriched quartz occurs at the Bilyan-Tau (924–1780 ppm), Isyanbet-I (1125–1754 ppm) and Alasiya-II (1352 ppm) deposits, while quartz from the Fazly-Tau (332–465 ppm) and Tukan (563 ppm) deposits are poor in water. In quartz from the Muildy-Tamak deposit, the amount of water varies dramatically from low in barren quartz to high in ore-bearing quartz (479–1580 ppm). At other deposits,

the amount of $H_2O$ in barren and ore-bearing quartz differs less dramatically. Notably, the $H_2O$ content in ore-bearing quartz can be both higher and lower.

**Table 4.** Results of gas analysis of fluid inclusions in the veins of quartz from gold deposits of the Khudolaz area (ppm).

| Sample | Quartz Vein Type | Deposit | $N_2$ | $CO_2$ | $H_2O$ | $CH_4$ | HG | $CO_2/CH_4$ |
|---|---|---|---|---|---|---|---|---|
| $K_{22}$-20b | white barren | Muildy-Tamak | b.d.l. | 47.0 | 478.5 | 3.3 | 2.1 | 14.4 |
| $K_{22}$-20c | brownish ore-bearing | | b.d.l. | 41.5 | 1580 | 5.9 | 1.5 | 7.1 |
| $K_{22}$-23a | white barren | Isyanbet-I | b.d.l. | 7.9 | 1125 | 3.7 | 0.4 | 2.1 |
| $K_{22}$-23b | brownish ore-bearing | | b.d.l. | 45.2 | 1754 | 3.8 | 0.9 | 11.9 |
| $K_{22}$-29a | brownish ore-bearing | Alasiya-II | b.d.l. | 52.2 | 1352 | 3.8 | 0.7 | 13.7 |
| $K_{22}$-32b | grayish barren | Fazly-Tau | 0.1 | 11.3 | 465.3 | 1.9 | 0.6 | 5.9 |
| $K_{22}$-32c | brownish ore-bearing | | 0.1 | 24.1 | 331.8 | 1.3 | 1.2 | 18.4 |
| $K_{22}$-38a | brownish ore-bearing | Bilyan-Tau | b.d.l. | 29.4 | 924 | 3.8 | 1.3 | 7.7 |
| $K_{22}$-38b | white barren | | b.d.l. | 27.2 | 1780 | 6.2 | 0.6 | 4.4 |
| $D_2$-36a | white barren | Tukan | 0.19 | 34.4 | 563.1 | 0.7 | 1.0 | 49.2 |

Note: b.d.l.—below detection limit, HG—common content of $C_2H_4$, $C_2H_6$, $C_3H_6$ and $C_3H_8$.

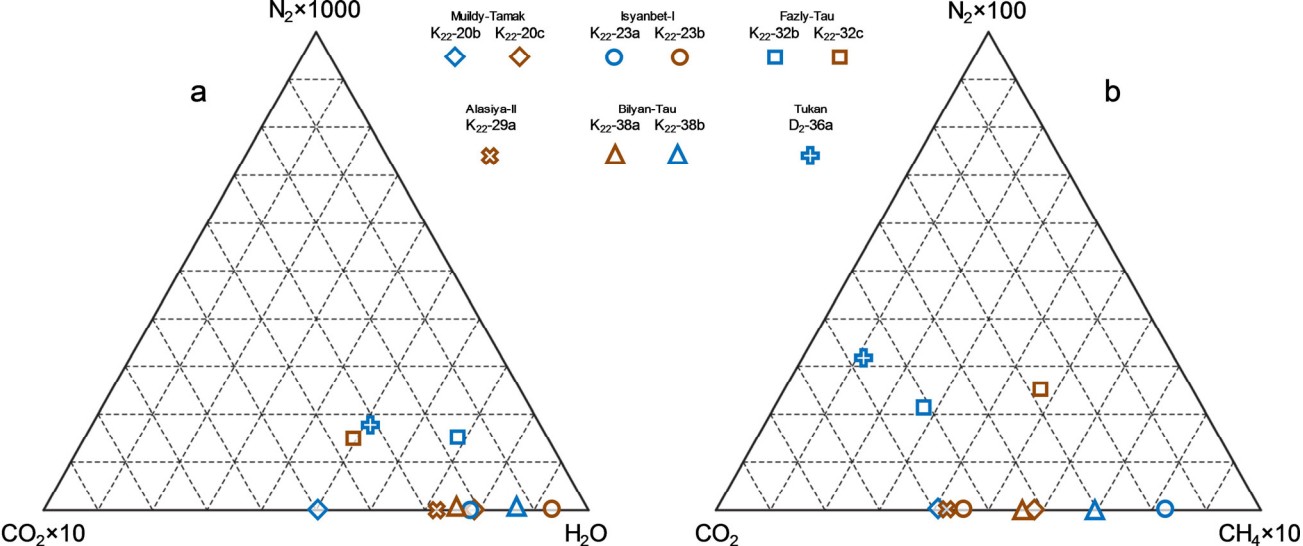

**Figure 11.** $N_2$–$CO_2$–$H_2O$ (**a**) and $N_2$–$CO_2$–$CH_4$ (**b**) ternary diagrams (in ppm) for fluid inclusions from quartz veins at gold deposits in the Khudolaz area. Brownish symbols—ore-bearing quartz. Blue symbols—barren quartz.

The $CO_2/CH_4$ value for quartz from the Tukan deposit is maximal (49.2), while it is several times lower in the other deposits (from 2.1 to 18.4). Notably, the $CO_2/CH_4$ value in this ore-bearing brownish quartz can be both higher and lower than that in barren quartz.

*4.6. Isotopy of Oxygen in Quartz*

Ten samples from quartz veins were selected to study the isotope composition of oxygen, i.e., two samples from Muildy-Tamak, Isyanbet-I, Fazly-Tau, and Bilyan-Tau deposits; one sample from Alasiya-II and Tukan deposits (Table 5). Values of $\delta^{18}O_{SMOW}$ in all samples fall into a narrow range of 16.1‰–18.2‰, while differences in the isotope composition of oxygen between ore-bearing (brownish) and barren (white) quartz are not detected. A relatively heavier isotope composition of oxygen is typical of quartz from the Isyanbet-I,

Alasiya-II and Tukan deposits (15.5‰–16.6‰), while a lighter composition is observed in quartz from the Muildy-Tamak, Fazly-Tau and Bilyan-Tau deposits (17.4‰–18.2‰).

**Table 5.** Oxygen isotope analyses of quartz veins from gold deposits in the Khudolaz area.

| Sample | Quartz Vein Type | Deposit | $\delta^{18}O$ ‰ in Quartz, V-SMOW | $\delta^{18}O$ ‰ in Fluid, Calculated |
|---|---|---|---|---|
| $K_{22}$-20b | white barren | Muildy-Tamak | 17.6 | 10.7 |
| $K_{22}$-20c | brownish ore-bearing | | 17.5 | 8.1 |
| $K_{22}$-23a | white barren | Isyanbet-I | 16.4 | 9.9 |
| $K_{22}$-23b | brownish ore-bearing | | 16.6 | 10.5 |
| $K_{22}$-29a | brownish ore-bearing | Alasiya-II | 15.5 | 9.9 |
| $K_{22}$-32b | grayish barren | Fazly-Tau | 17.5 | 8.1 |
| $K_{22}$-32c | brownish ore-bearing | | 17.5 | 7.7 |
| $K_{22}$-38a | brownish ore-bearing | Bilyan-Tau | 17.4 | 7.8 |
| $K_{22}$-38b | white barren | | 18.2 | 5.5 |
| $D_2$-36a | white barren | Tukan | 16.1 | 6.8 |

The isotope composition of oxygen in the fluid was estimated by the equation $\delta^{18}O_{quartz} - \delta^{18}O_{fluid} = 3.38 \times 10^6/T^2 - 3.40$ [34,35]. The mean values of homogenization temperatures of each sample recalculated for °K were used in the measurements. Calculated values of $\delta^{18}O_{fluid}$ vary from 5.5 to 10.7‰ (Table 5), falling into the range of magmatic and metamorphic fluids (Figure 12).

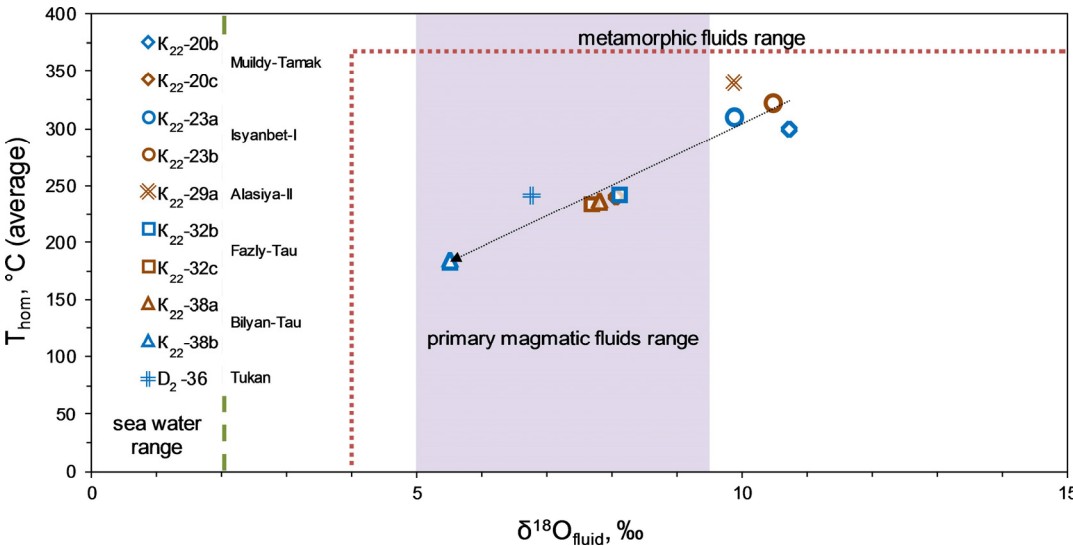

**Figure 12.** Calculated oxygen isotopic values ($\delta^{18}O_{fluid}$) vs. homogenization temperature ($T_{hom}$) plot [32] for fluid inclusions in quartz veins from gold deposits in the Khudolaz area. Brownish symbols—ore-bearing quartz. Blue symbols—barren quartz.

## 5. Discussion

### 5.1. Gold Deposits Distribution

The results of the geological study of six gold-quartz deposits in the Khudolaz area indicated their regular features. The deposits have a linear or sublinear structure with an NE strike; they are confined to intrusions of the Khudolaz complex and dikes of the Ulugurtau complex with late Devonian volcanogenic-sedimentary rocks. In most, but not in all deposits, bodies of both the Khudolaz (329–325 Ma) and Ulugurtau (321 ± 15 Ma) complexes were found, which suggests that either deposit could be a source for ore-bearing fluids. This allows us to limit the age range of the gold mineralization to 330–320 Ma. The

small sizes of the deposits and their large numbers correspond to those of the Khudolaz and Ulugurtau bodies. Hence, we can suggest that larger gold deposits can occur at a greater depth (>0.5 km) since geophysical data predict the presence of a large body of Khudolaz complex [24]. The reactivation of ancient and origination of new ore-controlling faults in this period of the South Urals evolution is linked to the process of oblique collision of the East European Paleocontinent and the Magnitogorsk island arc [19,24].

### 5.2. Origin of Ore-Bearing Fluids

#### 5.2.1. Stages of Mineralization

A wide diversity of ore mineral phases in gold-bearing quartz veins suggests various conditions of mineralization. Based on specific features of distribution and intergrowth of ore minerals, we define three major stages of mineral formation. The earliest stage was marked by the crystallization of early oxides and silicates of Fe and Ti (magnetite, ilmenite, titanite) that occur in early white barren quartz veins. Later (middle or main stage), a polymetallic association was formed (arsenopyrite, chalcopyrite, sphalerite, galena, pyrite, etc.) related to brownish quartz veins. At the latest stage (transient from hypogenic to hypergenic environmental settings), early-stage minerals were partly or completely replaced by low-temperature water-bearing minerals (goethite, rosasite, hydrocerussite, etc.). Using an arsenopyrite (As 30.34–32.09 atomic % at all) thermometer [36,37], the crystallization temperature of arsenopyrite in three deposits was estimated: Muildy-Tamak—341–394 °C, Fazly-Tau—351–441 °C, and Bilyan-Tau—322–526 °C. In the example of the Tukan deposit [23], we can suggest that native gold crystallized at the middle stage of mineral formation and, in smaller amounts, at the latest stage.

The range of homogenization temperature of fluid inclusions in quartz veins (164–354 °C) overlaps that of calculated formation temperatures of early chlorite (pycnochlorite) (170–276 °C, calculated using a geothermometer after [38]) intergrown with titanite. The formation temperature of late chlorite (ripidolite) associated with goethite is calculated to range from 135 to 149 °C. Thus, we limit the temperature of the main mineralization stage to >160 °C and the temperature of the late stage to <150 °C.

#### 5.2.2. Geochemical Evolution of Quartz Veins

As already mentioned, the geological setting of gold-bearing quartz veins in all deposits is similar. Quartz veins are confined to NE-trending faults near to boundaries of small intrusions with host rocks. We believe that the geochemical variations, as well as the mineralogical ones of quartz veins in the different deposits, are associated with three main factors: (1) composition of host rocks; (2) composition of embedded intrusions and (3) the degree of hydrothermal reworking of both intrusive and host rocks. The results of the geochemical study validated a higher productivity of brownish quartz not only for Au but also for other metals (chalcogen, ferrous and non-ferrous). At the same time, the concentration of gold does not correlate with the amount of sulfides and sulfoarsenides. The above data on the geochemistry of quartz veins, microthermometry and gas analysis indicate that the highest Au concentration (≥0.5 ppm) are typical of the following veins: (1) with temperatures of inclusions homogenization in the range of 230–330 °C and fluid salinity in the range of 8–12 wt.% $NaCl_{eq.}$; (2) coexisting with fluids rich in $H_2O$ and $CH_4$; and (3) relatively depleted in REE (except for veins of the Tukan deposit rich in REE due to the presence of calcite).

The quartz veins of the Tukan deposit are rich in REE and show a strong positive Eu-anomaly (Eu/Eu* = 11.3–11.5), which indicates a major amount of calcite admixture. Calcite from carbonate-quartz veins of gold deposits typically shows a positive Eu-anomaly [39,40] resulting from the change in redox conditions. It had been previously defined that the occurrence of native gold in the Tukan deposit was confined to the occurrence of calcite in the hydrothermal system [22]. It was likely to mark the change in oxidizing conditions to reducing ones, which led to the precipitation of gold. Notably, calcite is found in each studied deposit, but it is distributed irregularly.

The absence of Ce-anomaly can be explained by a weakly oxidized or neutral environment during the formation of quartz veins and a lack of impact of meteoric water on the fluid [39,41–43]. The U/Th ratio range (0.1–1.1) in quartz veins also indicates an oxidized and neutral environment [44]. The Y/Ho value is applied to interpret sources of ore-bearing fluids [45,46]. The Y/Ho ratio in quartz veins from all of the studied deposits varies from 29 to 41, which is overlapped by the range of Y/Ho ratios in rocks of the Khudolaz complex (23–45) [47]. Meanwhile, the Y/Ho ratio in the host sandstones and tuff sandstones of the Biyagoda and Zilair formations ranges from 24 to 32. The origin of distinct Tm-anomalies in quartz veins from the Muildy-Tamak, Isyanbet-I and Alasiya-II deposits is still unclear. This may be caused by the changes in redox condition that catalyzed unusual fractioning of heavy REE (HREE) since Tm is considered more volatile compared to other HREE [48].

### 5.2.3. Sources of Ore-Bearing Fluids

The results of the fluid inclusions studies indicate that gold-bearing and barren quartz veins at the small-sized deposits of the Khudolaz area were formed from weak to moderate saline (4.2–12 wt.% $NaCl_{eq.}$) aqueous chloride K-Na-bearing fluids at moderate temperatures (165–354 °C). No critical differences between ore-bearing and barren quartz veins were defined. At different deposits, the salinity and homogenization temperature of fluid inclusions in ore-bearing quartz can be both lower and higher than those in barren quartz. On the salinity vs. homogenization temperature plot, points of the studied deposits lie between calculated isobars of 10 and 200 bars (Figure 13a), corresponding to the epizonal class of deposits (depth < 6 km) [6]. The fluids contained $CO_2$ (8–52 ppm), $H_2O$ (924–1780 ppm), $CH_4$ (0.7–6.2 ppm) and a minor amount of heavy hydrocarbons (0.4–1.5 ppm) and $N_2$ (0.0n–0.19 ppm). The $\delta^{18}O_{SMOW}$ values in all the samples of quartz veins are in a narrow range of 16.1‰ to 18.2‰. All this testifies to the single source of ore-bearing and barren quartz and to an impulse mode of its injection, which predetermined the occurrence of mottled and banded (brownish-white) textures.

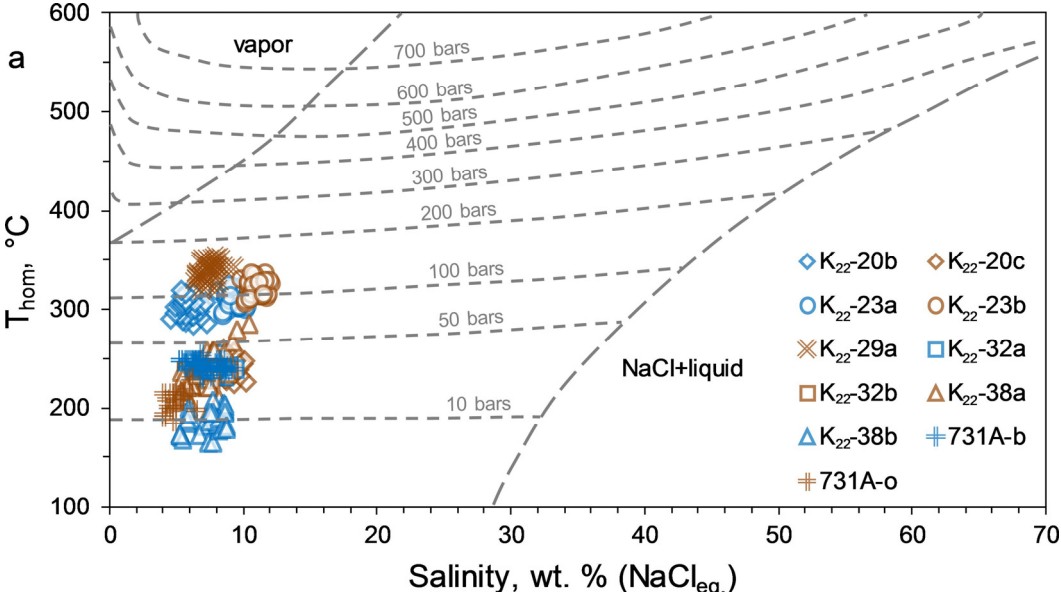

**Figure 13.** *Cont.*

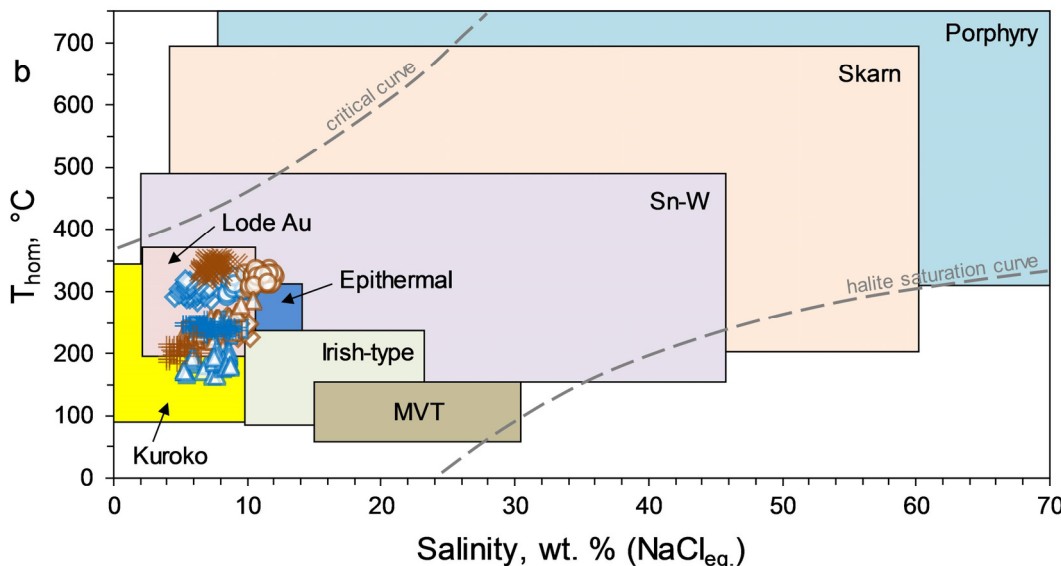

**Figure 13.** Homogenization temperature ($T_{hom}$) vs. salinity plots for fluid inclusions in quartz veins from the gold deposits in the Khudolaz area according to [49,50] (**a**) and to [51] (**b**). Brownish symbols—ore-bearing quartz. Blue symbols—barren quartz. MVT = Mississippi Valley-type Zn–Pb–F–Ba deposits.

Coexisting single-phase gas, fluid and more concentrated two-phase inclusions indicate the heterogenization of the fluid [52], which can be a reason for gold precipitation in quartz veins [53]. The prevalence of $CO_2$ among volatile components in fluids can be caused by the thermal impact of intrusions on sequences of limestone-bearing host rocks. The fluid from the Tukan deposit is specifically more reduced compared to fluids from other deposits. This is evidenced by its elevated $CO_2/CH_4$ ratio ($CO_2/CH_4 = 49$, in contrast to $CO_2/CH_4 = 2–18$ in other deposits).

The $\delta^{18}O_{fluid}$ range (5.5‰–10.7‰) nearly corresponds with etalon values of a magmatic fluid (5‰–9‰) (Figure 12); there are no clear signs of meteoric water impact. It is often difficult to differentiate magmatic and metamorphic fluids in their composition and PT parameters. The typical metamorphogenic fluid is considered to be rich in $CO_2$ and shows a relatively weak salinity [6,54,55], while quartz veins produced by this fluid commonly display negative Eu-anomalies and low REE values ($\Sigma$REE < 0.n ppm) [42,56–58]. Fluids of the studied minor deposits show weak to moderate salinity, low to moderate $CO_2$ content, neutral or positive Eu anomaly and moderate enrichment in REE ($\sum$REE $\geq$ 0.8 ppm). Thus, a significant influence of the magmatogenic fluid is traced. However, the obvious differences in the mineralogy and geochemistry of the studied deposits are probably due to the difference in the composition of the host rocks, which suggests the participation of fluid obtained during dehydration. In addition, it is doubtful that thin dikes could have produced sufficient fluid to form commercial gold mineralization. Therefore, we assume the formation of deposits is due to the reaction of the magmatogenic fluid with host rocks but without the participation of meteoric waters. Ranges of salinity and homogenization temperatures of fluid inclusions in quartz veins in the studied deposits are generally typical of lode gold fields (Figure 13b).

The difference in compositions of intrusive rocks and lithology of host rocks was the main factor in predetermining the mineralogical and geochemical diversity of quartz veins and compositional variations of gas in fluids. We consider intrusions of the Khudolaz and Ulugurtau complexes as the main source of gold. We believe that the gold was transported as bisulfide complexes (e.g., $Au(HS)_2^-$), as evidenced by the mesothermal regime of fluids and widespread sulfides in quartz veins [55,59]. Hydrothermally altered rocks of the Khudolaz and Ulugurtau intrusions with sulfide mineralization can be considered the main sulfur sources [25]. The geological structure of small-sized gold deposits of the

Khudolaz area, as well as the salinity and PT parameters of ore-bearing fluids, are typical of orogenic gold deposits [6]. Note the participation of magmatogenic fluid does not contradict the orogenic-type gold deposits [6,19]. The above-acquired dataset makes it possible to attribute the small-sized gold-bearing deposits of the Khudolaz area to the orogenic type [6,7,60,61].

## 6. Conclusions

A complex study of six minor gold deposits in the Khudolaz area of the South Urals was conducted. The current research provides the geological description, petrography of host rocks, mineralogy, geochemistry and oxygen isotopy of quartz veins, microthermometry of fluid inclusions in quartz and gas chromatography. Milky-white, mostly barren quartz veins and brownish ore-bearing quartz with native gold and various sulfides, sulfoarsenides, oxides, hydroxides, hydrosulfides and hydrocarbonates were identified. Bulk gold that occurs in free form or as intergrowths with sulfides was precipitated at temperatures of 230–330 °C from a weakly to moderately saline (8–12 wt.% NaCl-eq.) $H_2O$–$CO_2$–$CH_4$-bearing fluid, when weakly oxidized or near-neutral conditions, changed to reducing ones. Similar geological settings of ores in all deposits, a stable salinity range of fluids along with low hydrocarbon and $N_2$ content, as well as a narrow range of $\delta^{18}O$ values, indicate a prevailing magmatogenic source with a certain influence of host rocks but without the influence of meteoric waters. Intrusive bodies of the Khudolaz (325–329 Ma, U-Pb) and Ulugurtau (321 $\pm$ 15 Ma, Sm-Nd) complexes were the main source of the ore-bearing fluid. Based on the presented data, the studied deposits were attributed to the epizonal orogenic type. This study shows that the formation of lode gold deposits is possible without the participation of granite massifs.

**Author Contributions:** Conceptualization, I.R.R.; methodology, I.R.R., N.N.A., A.A.S. and S.N.S.; software, I.R.R. and A.A.S.; validation, I.R.R., N.N.A. and S.N.S.; formal analysis, I.R.R.; investigation, I.R.R., N.N.A. and A.A.S.; resources, I.R.R., N.N.A. and S.N.S.; data curation, I.R.R., N.N.A., A.A.S. and S.N.S.; writing—I.R.R. and A.A.S.; writing—review and editing, I.R.R. and N.N.A.; visualization, I.R.R.; supervision, I.R.R.; project administration, I.R.R.; funding acquisition, I.R.R. All authors have read and agreed to the published version of the manuscript.

**Funding:** This research was funded by Russian Science Foundation, grant number 22-77-10049.

**Conflicts of Interest:** The authors declare no conflict of interest.

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
