# Peer review of "Origin and Evolution of Ore-Forming Fluids at the Small-Sized Gold Deposits in the Khudolaz Area, Southern Urals"

_minerals, doi:10.3390/min13060781_

Round 1
Reviewer 1 Report
Review of minerals-2419373:
Title: Origin and Evolution of Ore-Forming Fluids at the Minor Gold Deposits in the Khudolaz Area, Southern Urals
Authors: Rakhimov et al.
The submitted manuscript discusses origin and sources of fluids producing a number of small-sized gold deposits in Khudolaz Trough of Southern Urals. The authors applied an impressive number of analytical methods to substantiate their findings including ICP-MS, SEM-EDS, microthermometry of fluid inclusions, gas chromatography, and oxygen isotopy. The presentation of article is good. Obtained results indicated that mineral formation in all the studied deposits occurred under similar mesothermal conditions at medium-low fluid salinity, although the mineralogy differs. The authors believe that the deposition of native gold occurred when the redox conditions of the fluid changed, but the reason for this is not very clear.
There is some information that is of interest and I am in favour of making studies such as this available to a wide readership. So, there is information here worth publishing- but the ms requires some moderate revision beforehand. This review makes some recommendations for changes which I deem essential prior to resubmission.
1. The term "minor" used by the authors in the manuscript to denote the insignificant resource potential of deposits is poor. It will be clearer to use the term "small-sized". the same applies to "country/wall rock" (change to "host rock")
2. The manuscript lacks at least schematic maps of studied deposits
3. The authors need to carefully edit the mineral composition and probably remove controversial indefinite phases (for example, Fe-S-O and Cd-Zn-S-O phases). Similar results have often obtained when analyzing a mixture of oxidized and non-oxidized sulfides by BSE.
4. I would not recommend the use of mineral thermometers in this study. Quartz veins are intensive oxidized and measured temperatures may show false values. In the work, high-level studies of fluid inclusions were carried out. This is what should be limited.
5. Section 5.2.1. Does it look strange that early iron oxide phases crystallized simultaneously with sulfides? Can it be worth distinguishing the early pre-ore stage with milky-white quartz, the main ore stage with sulfides, and the post-ore hypergene stage? It is not clear how gold could crystallize at the late hypergene stage.
6. In tables and on some Figs, authors need to add deposit labels, not just sample numbers.
7. Intrusion-related gold deposits are a product of local-scale fluids derived from a cooling pluton. This is different from orogenic deposits that are considered to result from crustal-scale fluids derived through metamorphic dehydration (Groves et al., 1998; Stuwe, 1998). Thus evidence of a magmatic source of the fluid is not clear. It is possible that the fluid separation from host rocks was initiated by intrusions, but then the fluid is rather orogenic. This is in complete agreement with all data in manuscript.
See some detailed notes in the annotated file

I am not a native English speaker and it is difficult for me to give a qualified assessment of the quality of English. I tried to make some changes in the language style, but I still recommend carefully editing the manuscript again.
Reviewer 2 Report
Please refer to the attachment for comments.

The native English speaker coauthor is responsible to polish the English to make the manuscript more academic, many of the words used in the paper are inappropriate, such as “vein quartz”, “gold quartz deposits”, “Gas analysis” and so on.
Reviewer 3 Report
The work is well organised and structured. The subject matter could be of interest to readers and researchers in this field.
However, the following observations are made, which the authors should consider in order to improve the work:
1. Abstract: The authors have to state the objective they intend to achieve with this work.
2. Abstract: The authors should state what methods they have used to develop this work.
3. Abstract: It is necessary to highlight the novelty of this work.
4. Abstract: Lines 15 and 16: the authors write: "...which is due to different composition of country rocks..."; the term "country rock" could be changed to " host rock"?
5. Authors are encouraged to add a paragraph at the end of the abstract highlighting how the results obtained could be applied in the field of science.
6. Figure 1. The authors have to indicate the study area on the geological map, as well as the location of the sampling points.
7. The authors should add field work such as geological survey and sampling to the "Methods" section. In addition, the total number of samples taken per area of interest should be mentioned. Please use a table to summarise the number of samples, type of analysis and locality.
8. Lines 565 to 567. The authors provide a categorical criterion in their discussion, but it is certain that no petrographic studies have been carried out to define the following issues:
a. It appears that the petrographic study belongs to earlier researchers.
b. Character of the alterations produced in the host rocks due to the intrusion of mafic and ultramafic rocks.
c. Did the hydrothermal fluids affect both the host rocks and the intrusive rocks equally?
d. In the case of the other rocks, such as sandstones, what is the relationship between them and the mineralisation fluid?
e. Was the study limited to quartz veins only?
f. Was the investigation carried out on the basis of previously stored hand samples or was geological survey work carried out?
g. The samples studied seem to come from the surface parts of former mine workings, which significantly reduces the possibilities of studying the deposits, as no samples from the lower levels are available. Therefore, the conclusions drawn by the authors may be extremely limited.
Minor editing of English language required
Round 2
Reviewer 3 Report
The authors have correctly answered the questions formulated by the Reviewer. They have also made the corrections indicated.
The reviewer considers that the paper now meets the requirements for publication in the Minerals journal.
I take this opportunity to congratulate the authors and wish them success in their future work.